# Noninvasive detection of audiovisual superadditivity in rat brain by miniaturized SERF magnetometer

**Kexun Tang[1], Guanzhong Lu[1], Peixin Zhang[1], Hao Wang[1], Zixuan Wang[1], Jia Yao[2], Yi Ruan[1]\*, Qiang Lin[3]\***

1 Key Laboratory of Quantum Precision Measurement of Zhejiang Province, School of Physics, Zhejiang University of Technology, Hangzhou, China, 2 Department of Breast Surgery, the First Affiliated Hospital of Zhejiang University, School of Medicine, Zhejiang University, Hangzhou, China, 3 State Key Laboratory of Ocean Sensing & Institute of Quantum Sensing & School of Physics, Zhejiang University, Hangzhou, China.

\* yiruan@zjut.edu.cn (YR); qlin@zju.edu.cn (QL)

## Abstract

The neural mechanisms of multisensory integration, particularly the superadditive response where combined sensory inputs elicit neural activity exceeding the sum of unimodal responses, remain a central frontier in perceptual and cognitive neuroscience. Utilizing a high sensitivity spin exchange relaxation free atomic magnetometer (SERF AM), we noninvasively record event related magnetic fields (ERMFs) in anesthetized rats during audiovisual (AV) stimulation, demonstrating the first application of SERF technology in rodent multisensory research. Comparisons across auditory-only, visual-only, and AV conditions revealed superadditive enhancement of the M300 component, with amplitudes surpassing the linear sum of unimodal responses. This effect is modulated by sound frequencies and inter-stimulus intervals (ISIs), suggesting dual modulation: frequency dependent consistency and ISI dependent emergence enhancement. AV induced M300 delays may reflect increased processing time under multisensory stimulation. The observed pattern of early suppression and later enhancement is consistent with condition dependent modulation across different stages of audiovisual processing. The approach advances cross species mechanistic understanding and provides a scalable platform for future translational research in neurodevelopmental conditions.

## Introduction

Multisensory integration represents a fundamental neural mechanism through which the brain merges inputs from different sensory modalities to create unified perceptual experience. A defining characteristic of this process is the superadditive response, where neural activity evoked by simultaneous multisensory stimulation significantly exceeds the linear sum of unimodal responses [1–3]. This nonlinear neural gain

**Data availability statement:** All relevant data are within the manuscript and its Supporting Information files.

**Funding:** National Natural Science Foundation of China (U20A20219, 61805213); Zhejiang Provincial Natural Science Foundation of China under Grant (LGF20C050001, LD22F050003). The funders played a significant role in this study by contributing to the study design, providing essential experimental equipment, and decision to publish and the preparation of the manuscript.

**Competing interests:** The authors have declared that no competing interests exist.

underlies enhanced perception, attentional allocation, and decision-making efficiency. The superadditive response has been widely observed across species ranging from rodents to humans using electrophysiological techniques such as event-related potentials (ERPs) and local field potentials (LFPs). These effects have been reported in the parietal cortex during multisensory decision-making tasks, in auditory cortex under context-dependent auditory inputs, and in excitatory-inhibitory dynamics of the primary auditory cortex [4–6].These electrophysiological findings have elucidated its role in early integration and cognitive modulation. Despite extensive electrical evidence, magnetic characterization of superadditivity in animal remains scarce. Compared with electrophysiological recordings, magnetic measurements offer several advantages: they are inherently non-contact and avoid invasive electrode implantation. Moreover, magnetic fields preserve vectorial information about current flow direction, enabling the analysis of neural signal orientation, which is lost in conventional electric potential recordings. These advantages render SERF AM, a highly sensitive and miniaturized magnetometer, particularly suitable for studying fine-scale neural dynamics in vivo with minimal interference.

Traditionally, multisensory timing has been explored with ERPs, which capture time-locked electrophysiological responses to sensory input [7,8]. Moreover, ERPs waveforms can be elicited using passive single-stimulus paradigms [9]. ERPs such as the auditory N1-P2 complex (100–200 ms latency) and the later P3 component (300–500 ms) respectively index early sensory gating and attentional shifts, as well as cognitive updating [10–12]. However, ERPs spatial resolution is limited by skull impedance and volume conduction. In contrast, ERMFs, which measure magnetic fluctuations generated by intracellular currents, offer millimeter-scale spatial resolution for cortical source localization [13,14]. ERMFs studies in humans have characterized auditory cortex like M100 component (≈100 ms latency), including its frequency-tuning properties [15]. In our prior studies, a custom-built SERF AM system was employed to detect auditory P3 and visual N2-like components in rats, whose amplitudes were significantly modulated by stimulus frequency and ISI, mirroring time-dependent processing features seen in human ERP studies [16,17]. However, it remains unknown whether synchronized audiovisual stimulation can evoke superadditive magnetic responses in rat brain, or how such integration is dynamically modulated by parameters like frequencies and ISIs [18,19].

Expanding on our earlier findings, this study addresses these critical knowledge gaps by investigating whether synchronized audiovisual stimulation elicits superadditive M300 response in rats. Using a high-sensitivity SERF AM system, we recorded cortical ERMFs from anesthetized rats presented with auditory (A), visual (V), and audiovisual (AV) stimulation, with experimental conditions systematically varied across sound frequencies and ISIs By varying tone frequencies (2.3–8.3 kHz) and ISIs (3–9 s) in a passive single-stimulus paradigm and quantifying responses, with a superadditive response index (SRI), we demonstrate frequency- and time-dependent superadditive magnetic responses in rodents. These findings validate SERF AM as a powerful tool for neural measurement and open new avenues for cross-species studies of multisensory integration.

## Experimental methodology

### Animals preparation

This study strictly adhered to the Guide for the Care and Use of Laboratory Animals, and all procedures were approved by the Ethics Committee of Zhejiang University of Technology (Approval No. 20230703017). A total of 96 adult female BALB/c rats (body weight: ≈180 g) were used, obtained from the Laboratory Animal Center of Zhejiang Chinese Medical University (specific pathogen-free grade). The animals were housed in a temperature-controlled environment with a 12:12 hours light–dark cycle. Food and water were withheld for 12 hours prior to the experiment to stabilize metabolic conditions. Anesthesia was induced by intraperitoneal injection of sodium pentobarbital (40 mg/kg body weight), and respiratory rate and reflexes were continuously monitored to ensure adequate anesthetic depth. After stable anesthesia was achieved (≈10 minutes), each rat was secured on a non-magnetic foam vibration-isolation platform and placed inside a five-layer mu-metal magnetic shielding chamber. To minimize motion artifacts, the limbs were gently restrained using medical tape, and scalp hair over the skull was shaved to ensure optimal sensor proximity (approximately 1 mm above the scalp surface). At the end of the experiment, euthanasia was performed via intraperitoneal injection of an overdose of sodium pentobarbital (100 mg/kg), following the guidelines for euthanasia of laboratory animals, to ensure a humane and painless termination.

### SERF AM systems

The SERF AM operates using alkali metal vapor at high atomic number densities to achieve high sensitivity in detecting extremely weak magnetic fields. Under optical pumping, the net spin polarization P follows the Bloch equation:

$$\frac{dP}{dt} = \gamma P \times B + \frac{P_0 - P_z}{T_1}\hat{z} - \frac{P_x\hat{x} + P_y\hat{y}}{T_2},$$

(1)

In the steady-state limit and under the small-signal approximation($P_x, P_y \ll P_0$), a weak transverse magnetic field $B_x$ yields a spin polarization $P_x$:

$$P_x = \frac{sR\tau}{1 + \gamma^2 B_x^2\tau^2},$$

(2)

Here $\gamma$ is the gyromagnetic ratio, $T_1$ and $T_2$ are longitudinal and transverse relaxation times of the alkali atom, and $P_0$ is the steady-state spin polarization under optical pumping. R denotes the optical pumping rate, and $s$ represents the polarization of photon, and $\tau = (R + 1/T_2)$.

The sensitivity of conventional optical magnetometers is limited by spin-exchange collision-induced broadening [20].At low spin-exchange rates ($\Gamma_{SE} \ll \omega_L$), each collision introduces decoherence, leading to redistribution among the Zeeman sublevels $m_F$, while conserving the total atomic spin. However, in the SERF regime($\Gamma_{SE} \gg \omega_L$), the rapid interconversion between hyperfine states effectively suppresses spin ensembles precession between collisions, thereby averaging out the decoherence induced by spin-exchange interactions [21].

In SERF regime, a weak modulation magnetic field $B_{mod}t = B_{mod}cos(\omega t)$ is applied along the direction orthogonal to the light propagation axis. This modulation shifts the signal to a higher frequency, allowing lock-in detection to suppress 1/$f$ noise.

The resulting spin polarization $P_x$ oscillates at the modulation frequency, and its amplitude can be expressed as:

$$P_x = \frac{2sR\tau B_x\tau^2 sin\omega t}{1 + \gamma^2 B_x^2\tau^2}J_0\left(\frac{\gamma B_{mod}}{q(\mathbf{P})\omega}\right) J_1\left(\frac{\gamma B_{mod}}{q(\mathbf{P})\omega}\right),$$

(3)

Where $J_0$ and $J_1$ are Bessel functions of the first kind, $q(\mathbf{P})$ is the nuclear slowing down factor.

In this study, elliptically polarized light was used as both the pump and probe source simultaneously [22–24]. The light can be expressed as a superposition of circular polarization state $\hat{\sigma}_+$ and linear polarization state. The circular component primarily facilitates uniform spin polarization of alkali atoms, while the linear component enables detection of the magneto-optical rotation induced by spin-polarized atoms. In our experiment, the angle between the optic axes of the quarter wave plate and the polarizing beamsplitter is set to $\pi/8$.

As elliptically polarized light propagates through the alkali vapor cell, the $\hat{\sigma}_+$ components selectively excite $\Delta m_F = +1$ transitions. Under the Kramers-Kronig approximation, the difference in refractive index between the circular components becomes:

$$n_+ - n_- = \frac{1}{2}Re(\chi_+ - \chi_-) \approx \frac{N|\mu|^2 \Delta_z}{\hbar\epsilon_0 \left(\Delta_0^2 + \Gamma^2\right)},$$

(4)

Here, $\chi_\pm$ are the electromagnetic susceptibilities for $\hat{\sigma}_\pm$ components, $\Delta_0$ is the detuning from atomic resonance, $\Delta_z = \gamma B_x$ is the Zeeman splitting, and $\Gamma$ is the total linewidth. The dipole matrix element $\mu$ characterizes the strength of the atomic transition, and $N$ is the atomic number density.

This birefringence induces a Faraday rotation of the probe polarization, known as magneto-optical rotation (MOR). For probe path length l and wavelength $\lambda$, the rotation angle $\phi$ is:

$$\phi = \frac{\pi l}{\lambda}\left(n_+ - n_-\right) = \frac{\pi l N|\mu|^2}{\lambda\hbar\epsilon_0}\frac{\gamma B_x}{\left(\Delta_0^2 + \Gamma^2\right)}\left(\Delta_z \ll \Delta_0\right),$$

(5)

A commercially produced SERF AM (LCZ-01, Hangzhou Q-Mag Technology Co., Ltd.) was laboratory-adapted for the non-invasive recording of neural activity-induced ERMFs in anesthetized rats. The system's core comprises a frequency-stabilized laser (blue-detuned from the $^{87}Rb$ D1 line by approximately tens of GHz), designed to maximize atom-light coupling efficiency for high-fidelity signal acquisition. The output beam is modulated into elliptically polarized light through polarization optics and serves simultaneously as both pump and probe to continuously polarize the $^{87}Rb$ alkali vapor. The vapor cell itself is a miniaturized cubic enclosure (3×3×3 mm³) containing high-density enriched $^{87}Rb$ atoms, mixed with 760 Torr of nitrogen ($N_2$) and an appropriate amount of helium. These buffer and quenching gases help maintain the spin-polarized state and reduce radiation trapping. To avoid magnetic interference during the heating process, the vapor cell is inductively heated using a 1300 kHz alternating current. The heating structure comprises a boron nitride shell wound with ultra-fine, double-twisted enamel-coated copper wire, achieving a stable operational temperature of 160°C. At this temperature, the vapor density of $^{87}Rb$ reaches approximately $1.625 \times 10^{14}$ atoms/cm³, ensuring optimal SERF conditions. To detect the magnetic field–induced polarization rotation, the outgoing probe beam passes through a polarization detection module consisting of a half-wave plate, a Wollaston prism, and a balanced photodetector. After traversing the spin-polarized vapor, the probe experiences a slight magneto-optical rotation, which is transformed into an intensity difference between two orthogonal linear polarization components by the prism. This differential signal is captured by the photodetector. Under small magnetic fields, the rotation angle $\phi$ is approximately linear with respect to the applied transverse field $B_x$, resulting in the output voltage:

$$V_{sig} = GI_0\sin\left(2\phi\right) \approx 2GI_0\phi = KB_x,$$

(6)

where $K = 2GI_0\frac{\pi l N|\mu|^2}{\lambda\hbar\epsilon_0}\frac{\gamma}{(\Delta_0^2+\Gamma^2)}$ is the conversion gain, dependent on the optical properties and atomic parameters. In the absence of field-induced rotation, the polarization axis aligns with the analyzer, and detector outputs are balanced. Small

changes in magnetic field induce a proportional imbalance, enabling precise measurement of $\phi$ via lock-in amplification and subsequent inference of $B_x$.

To maintain a magnetically quiet environment, the magnetometer is enclosed within a five-layer mu-metal shielding assembly that offers high magnetic permeability and effectively blocks geomagnetic and environmental stray fields. The residual field inside is reduced below 10 nT. Surrounding the vapor cell, three-axis Helmholtz coils are installed to provide tunable magnetic fields in the range of ±30 nT. These coils are dynamically controlled via a PID feedback loop that continuously adjusts the compensation current to nullify remaining field components, thereby maintaining stable weak-field conditions required for optimal SERF operation. This active compensation system ensures stable field conditions within the SERF regime over long experimental durations.

As shown in Fig 1, the sensor head achieves a sensitivity of $20fT/\sqrt{Hz}$ and a bandwidth of 1–80 Hz, comparable to the $15fT/\sqrt{Hz}$ performance reported by Quspin Inc. The design is compact and highly integrated, as shown in Fig 2(a)-(b), with a sensor width of only 25 mm and a vapor cell positioned just 4 mm from the external housing. This enables near-zero distance placement above the rat skull, significantly improving signal strength over conventional superconducting quantum interference device (SQUID) systems. Moreover, the system is lightweight, power-efficient, and compatible with wearable applications, offering a new technological pathway for neural magnetic field detection.

## Experimental setup and procedure

The experimental system comprised a commercially developed SERF AM, a non-magnetic vibration-isolation foam platform, multisensory stimulus sources, and a TTL-based timing control module (Fig 2(c)-(f)). To eliminate motion artifacts introduced by active discrimination and behavioral feedback in traditional oddball paradigms, we adopted a passive single-stimulus paradigm [25,26] in which each trial presented only one type of stimulus—either visual (V), auditory (A), or synchronous audiovisual (AV)—without requiring any behavioral response or training from the animal. This paradigm relies on the unpredictability of stimulus occurrence to elicit passive sensory responses, which we previously demonstrated to reliably evoke ERMF components such as P300 and N2-like waves in anesthetized rats [16,17]. The magnetometer was positioned over the right dorsal skull of the rat to record ERMF signals from the left cerebral cortex, where sensory inputs from the contralateral (left) eye and ear are primarily processed [27,28]. Following the guidance [29], the

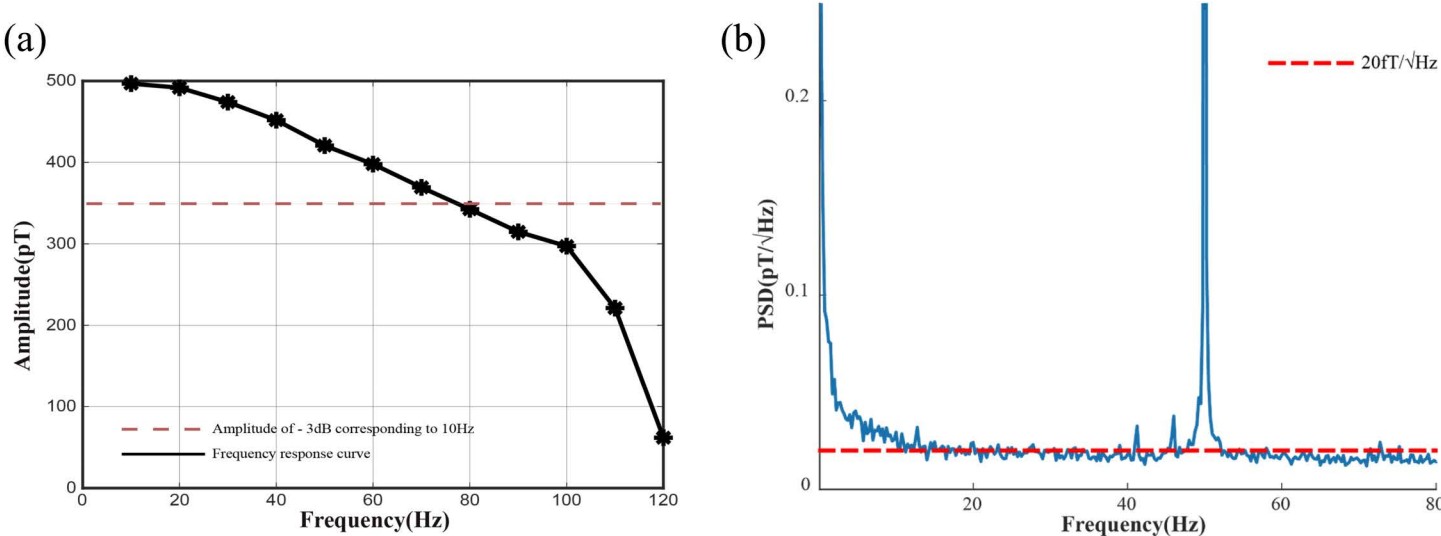

**Fig 1. (a) Frequency response curve with the amplitude beginning at 500 pT.** (b) Sensitivity of SERF AM.

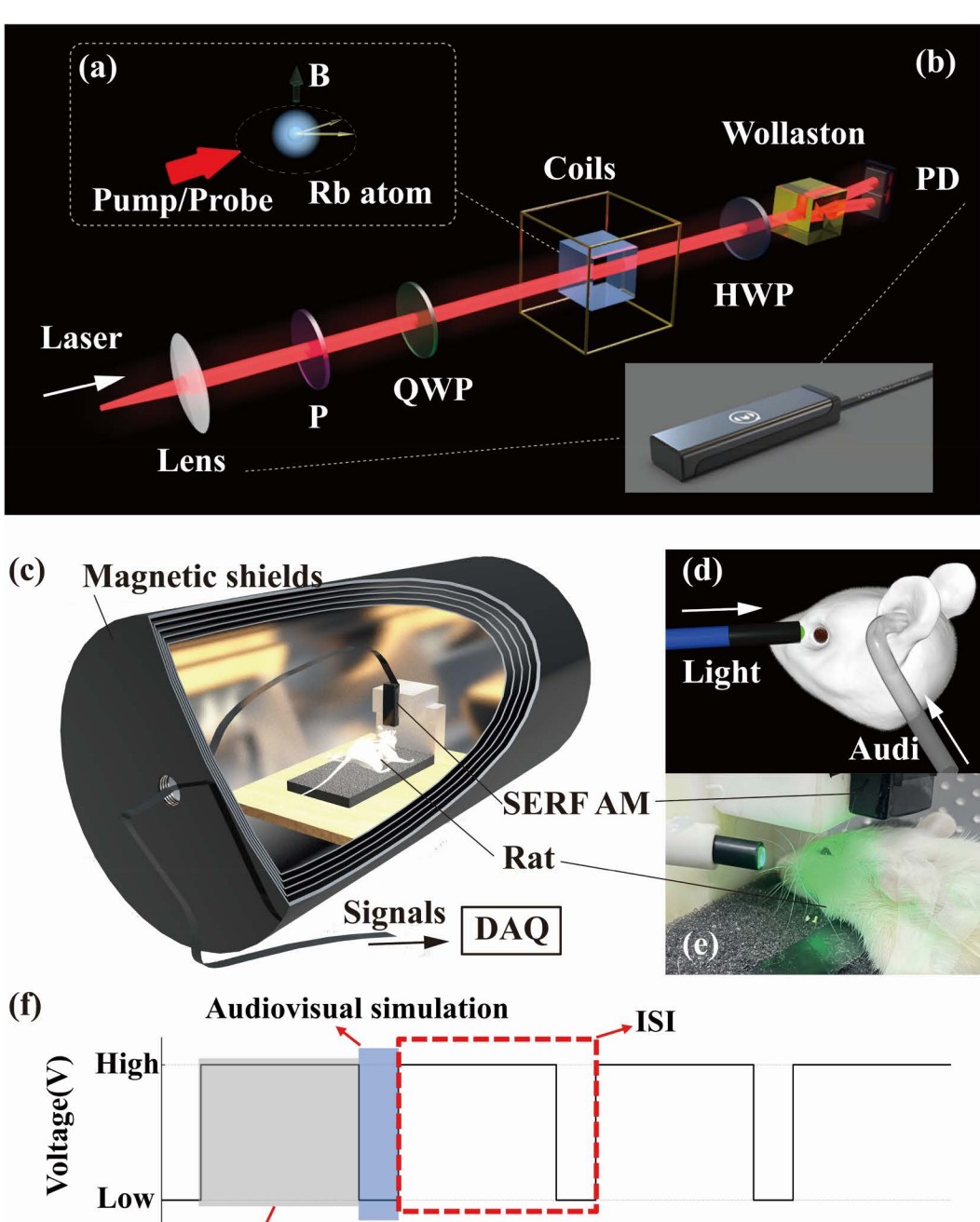

**Fig 2. (a) principle of the SERF AM system: Rb atoms are optically pumped and probed by elliptically polarized light, with spin polarization precessing under an external magnetic field B.** (b) Structure of the SERF AM system, comprising the laser beam path with optical components: lens, polarizer (P), quarter-wave plate (QWP), atomic vapor cell surrounded by Helmholtz coils, half-wave plate (HWP), Wollaston prism, and balanced photodetector (PD). The commercially developed SERF AM used in this study is shown below, featuring miniaturized sensor head. The figure has been authorized by Hangzhou Q-Mag Technology Co., Ltd. (c) Schematic of the experimental setup inside the five-layer mu-metal magnetic shields, where the anesthetized rat is placed on a vibration-isolated foam platform. The SERF AM is positioned above the anesthetized rat, and the signal is transmitted via cables to an external data acquisition (DAQ) system. (d) Illustration of audiovisual stimulation: a green LED light is delivered to the left eye via optical fiber, and auditory stimuli are transmitted via a soft sound tube to the left ear. (e) Photograph of the actual experiment showing placement of optical and acoustic channels. (f) Timing diagram of audiovisual stimulation paradigm is shown upper right. Audiovisual stimuli were delivered during the low-voltage phase, followed by a silent inter-stimulus interval (ISI). The paradigm used a passive single-stimulus design to induce passive sensory responses.

placement of the magnetometer was guided by the standard landmark of one-third the distance from the cranial vertex (Cz) to the external auditory meatus. This positioning targets cortical regions known to generate N2 and P300 components to maximize the signal-to-noise ratio, allowing simultaneous coverage of parietal areas and frontal regions, thus maximizing detection of audiovisual integration responses. And the Z-axis of the sensor was oriented perpendicular to the cortical surface, with the sensing element located approximately 1 mm above the scalp. To ensure stable stimulus delivery and minimize mechanical disturbances, the optical fiber and acoustic tubing were independently mounted on separate support structures. The visual stimulus was a 510 nm narrowband LED coupled into an optical fiber and delivered approximately 1 cm in front of the rat's left eye. The optical output at the fiber tip was 5 mW. Auditory stimuli were transmitted through a 5 cm long, 1.5 mm inner-diameter flexible sealed tube connected to a speaker and inserted into the left ear of the rat, sealed with medical adhesive tape. All auditory signals were pre-calibrated at the tube outlet using a sound level meter and maintained at 90 dB SPL to obtain robust evoked responses under pentobarbital anesthesia [30,31]. To rule out stimulus and equipment artifacts, recording using the identical stimulation timing and hardware configuration without rat was performed. Under this condition, the traces remained close to the baseline and showed no time-locked components.

We employed a sequence of trials to compare ERMF responses elicited by AV stimulation with those evoked under unimodal A, V, and Null (no-stimulus) conditions, enabling the analysis of the neural characteristics of audiovisual integration. To test for superadditive effects while minimizing potential confounds from temporal expectancy and rhythmic stimulation, we followed the approach of Zumer and Teder comparing (AV + Null) against (A + V) [32,33]. In our experiment, the consistently near zero ERMF responses in the Null condition indicate that the optical fiber and acoustic tube do not introduce periodic magnetic signals, confirming its suitability as a non-interacting baseline for multisensory interaction effects. In this situation, AV + Null was numerically equivalent to AV within the resolution of the present measurements; therefore, the descriptive SRI was computed using AV for simplicity. Auditory stimuli were prepared at four distinct frequencies (2.3, 4.3, 6.3, and 8.3 kHz) and delivered through the speaker system. The stimulus output was modulated by Transistor-Transistor Logic (TTL) pulse width, and each auditory frequency was tested at four ISI (3, 5, 7, or 9 s), yielding a total of 16 frequency–ISI combinations. Stimulus timing was controlled by TTL pulses: a high-level signal (5 V) represented baseline silence, while a low-level signal (0 V) triggered the stimulus, with each trial lasting 1 second. A total of 96 adult female rats were used in this study. For each distinct combination of auditory frequency and inter-stimulus interval (ISI), six independent rats were assigned to receive auditory (A), visual (V), and audiovisual (AV) stimuli under fixed stimulus parameters. Within each test cycle using a fixed frequency–ISI combination, A, V, and AV trials were randomly interleaved to prevent order effects of multisensory neural responses. For each rat, a recording run consisted of approximately 120 stimulus repetitions. After artifact rejection, 100 trials typically remained and were averaged to yield a rat level ERMF waveform for that condition with a high signal-to-noise ratio. To ensure sufficient data quality while minimizing habituation or fatigue, the same rat could be recorded again under the same fixed parameters after several days for typically 2–3 runs. These repeat runs were performed to obtain a usable evoked waveform under a stable physiological state; runs with gross instability or excessive artifacts were discarded. For downstream statistical analyzes, one retained run per rat and condition was retained.

ERMF signals were recorded using a National Instruments USB-6281 acquisition system at a sampling rate of 10 kHz. The data processing pipeline included the following steps:

(1) Bandpass filtering: Stimulus onset and offset were identified, and data were segmented accordingly. A 0.1–40 Hz frequency domain FFT bandpass filter was applied to suppress baseline drift and high frequency noise. Frequency below 0.1 Hz and above 40 Hz were set to zero, and the filtered signal was reconstructed by inverse FFT. Stimulus epochs were then segmented according to trigger rising and falling edges.

(2) Baseline correction: A 200 ms pre-stimulus window was used to calculate the baseline mean, which was subtracted from the entire trial to calibrate the signal.

(3) Artifact rejection: Each epoch was first smoothed using a 100 ms moving average window. Trials were excluded if the smoothed signal exceeded a fix threshold of ±1 at any time whithin the epoch. This procedure removed large amplitude artifacts, which were mainly associated with motion related disturbances or unstable physiological state. This threshold was chosen based on prior calibration experiments and visual inspection of representative segments.

(4) Signal averaging and component quantification: Averaged waveforms were computed across trials. Following nomenclature commonly used in human ERP and ERMF studies, response components were labeled approximately by latency as M100 [80 ms-120 ms] and M300 [300 ms-550 ms]. The amplitude of each component was approximately quantified by locating at the peak extremum within a predefined latency window centered on its expected latency.

(5) SRI calculation: The Superadditive Response Index (SRI) was calculated as $SRI = (AV - (A + V))/(A + V)$, where AV, A, and V represents the signed peak amplitude under audiovisual, auditory and visual conditions, respectively. Because Null responses were consistently near zero in this experiment, this expression was numerically equivalent, for practical purposes, to using AV in place of AV+Null.

(6) Statistical analysis: Statistical analyses were performed using rat as the experimental unit. Two-way analysis of variance (ANOVA) was conducted to test the effects of stimulus frequency and ISI and the response variable was the M300 peak amplitude under the AV condition. Two-way ANOVA tests were performed, and the corresponding F- and p-values and Cohen's d values were reported.

## Results

To characterize the temporal dynamics of multisensory-evoked brain magnetic responses in rats, we first compared the averaged event-related magnetic field (ERMF) waveforms under visual (V), auditory (A), and audiovisual (AV) stimulation conditions (Fig 3).

All traces were computed by averaging the rat level waveforms across 6 rats per condition, temporally aligned to the onset of stimulus (0 ms). Under the V condition, most combinations of stimulus frequency and inter-stimulus interval (ISI) evoked a robust negative deflection peaking around 80–120 ms, consistent with the N1 component, which may correspond to the human N2 observed in visual ERPs.

In contrast, under AV condition the M100 component was markedly attenuated or even absent in most frequency–ISI combinations. Notable exceptions occurred only at specific combinations (e.g., 2.3 kHz at ISI = 5 s; 6.3 kHz at ISI = 7 s; 8.3 kHz at ISI = 5 s and 7 s), where an M100 was still observable. This observation may be consistent with existing theories of sensory channel competition and early selective inhibition, which suggests that under simultaneous audiovisual input, neural responses in some sensory channels are suppressed to reduce redundancy [34–36].

In the M300 window, the AV condition elicited larger amplitudes than either A or V alone in most stimulus combinations. Notably, M300 amplitudes under AV conditions generally exceeded the linear sum of A and V responses, indicating the presence of a superadditive integration effect. However, this M300 enhancement was not uniformly distributed across all parameter conditions. At the 6.3 kHz, M300 amplitudes were more sensitive to ISI, exhibiting enhancement only at longer intervals (7–9 s), whereas high (8.3 kHz) and low (2.3 kHz) frequencies elicited consistently responses across most ISI settings.

Latency analysis further revealed that M300 peaks under AV stimulation were delayed compared with those under A only conditions. This delay may indicate increased processing time under multisensory stimulation. One possible explanation is that integrating concurrent auditory and visual inputs requires additional coordination across sensory pathways. As has been suggested in prior human MEG and ERP studies of supporting the integration cost hypothesis, which posits that although multisensory input boosts neural gain, the requirement for cross-channel coordination, conflict suppression, and attentional shifts prolongs processing latency [35,37].

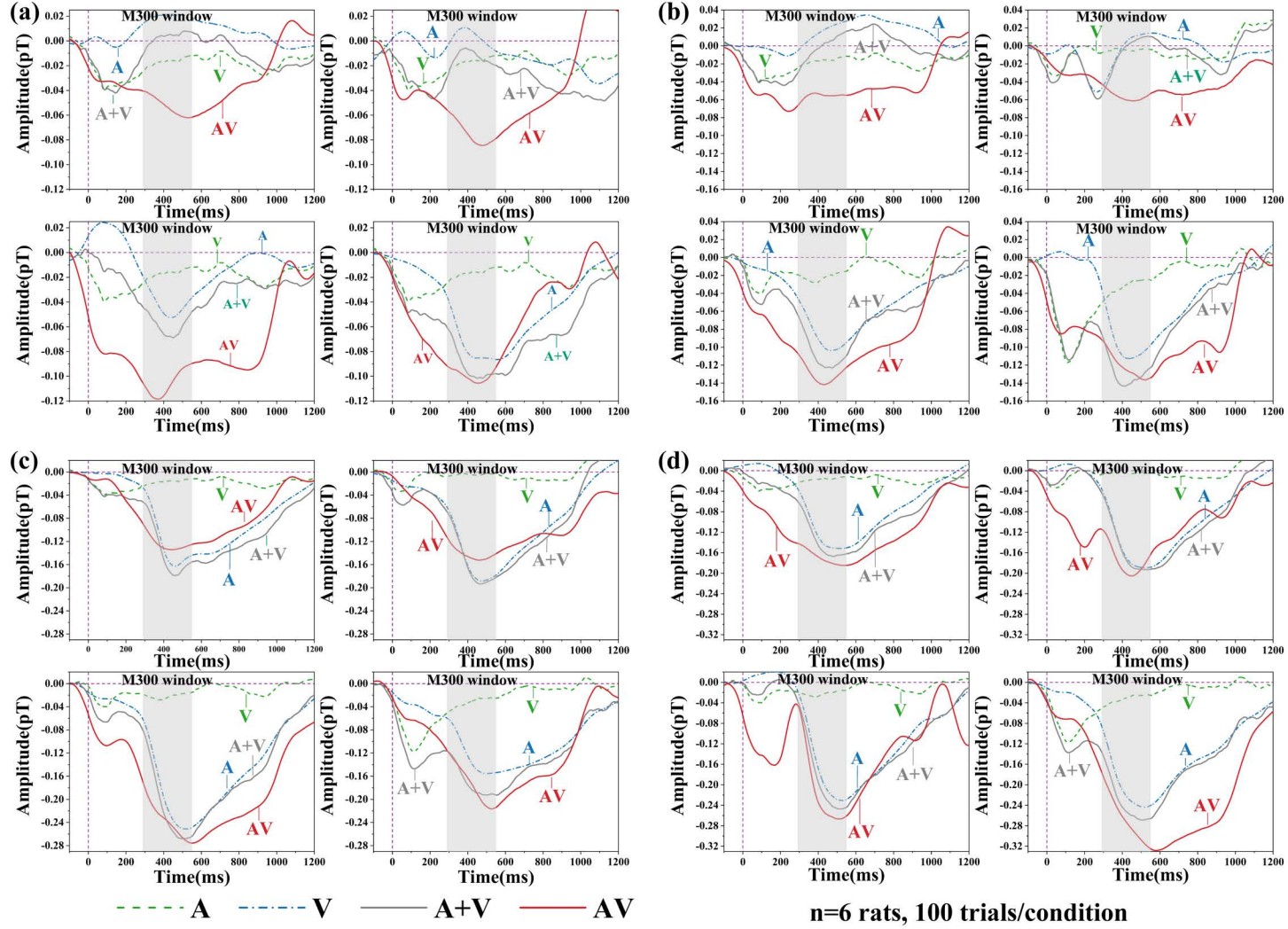

**Fig 3. (a)–(d) represent stimulation at four different auditory frequencies: (a) 2.3 kHz, (b) 4.3 kHz, (c) 6.3 kHz, and (d) 8.3 kHz.** Subplots from upper left to lower right correspond to four different ISI conditions: 3 s, 5 s, 7 s, and 9 s. The colored traces denote grand-averaged ERMFs across six independent rats under auditory (A, dashed blue), visual (V, dashed green), audiovisual (AV, solid red) conditions, and the linear sum of A and V (A+V, solid gray). The time axis is aligned at 0 ms corresponding to stimulus onset (vertical dashed purple line). The shaded gray region marks the M300 window.

To quantitatively assess the strength of audiovisual integration, we calculated the superadditive response index (SRI) under each stimulus condition: $SRI = \frac{AV-(A+V)}{A+V}$.

Where AV denotes the ERMF amplitudes under audiovisual stimulation, and A and V are amplitudes under unimodal conditions. The results are visualized in the heatmap shown in Fig 4, with corresponding absolute numeric values detailed in Table 1. These findings demonstrate that superadditive response strength is modulated by both auditory frequency and stimulus interval.

The data reveal pronounced nonlinear modulation of SRI. The heatmap and Table 1 are presented as descriptive summaries across the conditions. In the present dataset, the A+V remained positive in all tested conditions, no sign reversal occurred in the SRI calculation. The highest SRI values were observed at 2.3 kHz and 4.3 kHz under short ISIs, indicating nonlinear enhancement of the AV response beyond unimodal summation. In contrast, negative SRI values emerged under

## Superadditive Response Index (SRI) heatmap

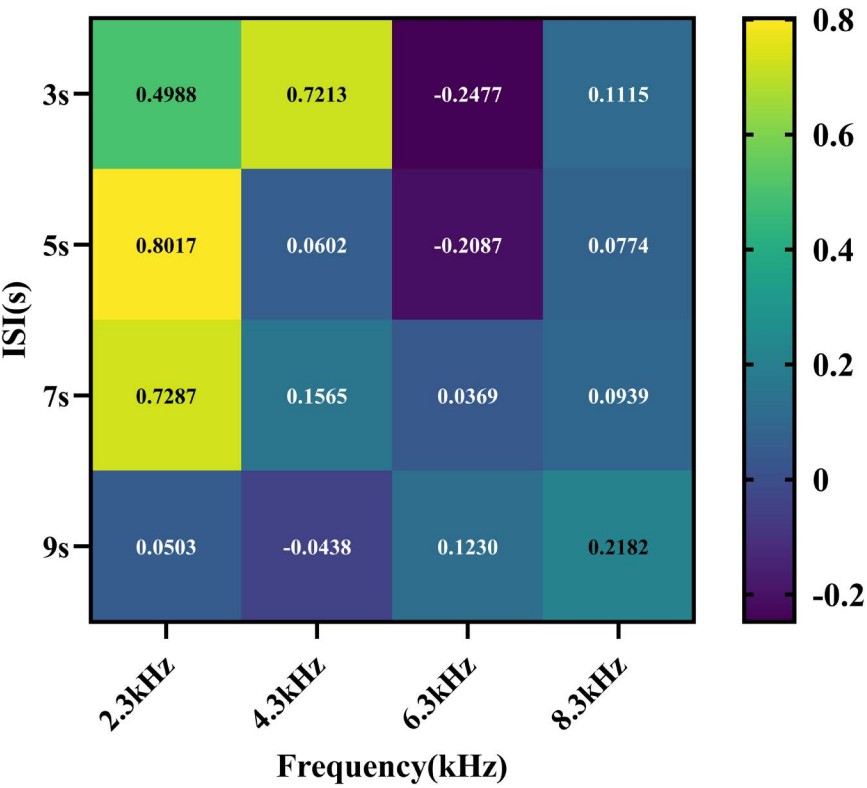

**Fig 4. Superadditive Response Index (SRI) heatmap across different auditory frequencies and ISIs.** The heatmap provides a descriptive summary of the SRI values under the 16 frequency-ISI combinations. SRI values greater than zero indicate superadditive integration. Light colors reflect stronger superadditive effects, while deep colors suggest weaker or even suppressive integration.

6.3 kHz under 3 s and 5 s, suggesting subadditive effects. As ISI increased, SRI values for 2.3 kHz and 4.3 kHz gradually declined, whereas those for 6.3 kHz reversed from negative to positive, peaking modestly at 9s. This ISI-dependent modulation highlights a possible interaction between stimulus interval and specific frequency. Across most conditions, shorter ISIs were associated with higher superadditive effects, suggesting that audiovisual integration may rely on frequency-dependent neural routing mechanisms, with each frequency range exhibiting a preferred temporal profile for effective cross-modal alignment. These findings suggest that while low-frequency stimuli benefit from short ISIs, certain mid-frequency inputs may require longer recovery periods to achieve effective integration. The overall response pattern underscores the complex interplay between auditory tuning properties and temporal expectancy mechanisms in shaping cross-modal gain.

These observations suggest that both stimulus frequency and temporal interval influence superadditive effects, though the underlying mechanisms remain to be clarified. One possibility is that different frequencies engage the auditory periphery and cortical representation with varying efficiency. Another is that ISI may modulate expectancy or novelty, which in turn could influence neural response deployment.

To quantitatively assess how auditory frequency and ISI jointly affect AV-induced M300 amplitude, a two-way ANOVA was conducted using both factors as independent variables. Normality of residuals was assessed using the Shapiro–Wilk test, which indicated some deviation from normality ($p = 0.0149$). Homogeneity of variance was assessed using Levene's test and was not significantly violated ($p = 0.573$). Given the balanced design, the ANOVA was considered reasonably

**Table 1. AV, A+V, SRI values under different combinations of frequencies and ISIs.**

| Frequency | ISI(S) | A+V/pT | AV/pT | SRI |
|---|---|---|---|---|
| **2.3 kHz** | 3 | 0.0415 | 0.0622 | 0.4988 |
| | 5 | 0.0471 | 0.0849 | 0.8017 |
| | 7 | 0.0685 | 0.1184 | 0.7287 |
| | 9 | 0.1007 | 0.1057 | 0.0503 |
| **4.3 kHz** | 3 | 0.0427 | 0.0735 | 0.7213 |
| | 5 | 0.0586 | 0.0624 | 0.0602 |
| | 7 | 0.1225 | 0.1417 | 0.1565 |
| | 9 | 0.1426 | 0.1364 | −0.0438 |
| **6.3 kHz** | 3 | 0.1779 | 0.1339 | −0.2477 |
| | 5 | 0.1918 | 0.1518 | −0.2087 |
| | 7 | 0.2664 | 0.2763 | 0.0369 |
| | 9 | 0.1921 | 0.2157 | 0.1230 |
| **8.3 kHz** | 3 | 0.1683 | 0.1871 | 0.1115 |
| | 5 | 0.1905 | 0.2053 | 0.0774 |
| | 7 | 0.2445 | 0.2675 | 0.0939 |
| | 9 | 0.2682 | 0.3267 | 0.2182 |

robust for the present analysis. The results showed statistically significant main effects of both auditory frequency ($F(3, 80) = 39.193$, $p = 7.95 \times 10^{-11}$) and ISI ($F(3, 80) = 13.939$, $p = 5.53 \times 10^{-6}$), while the interaction between these two factors was not significant ($F(9,80) = 0.654$, $p = 0.743$). These results indicate that both frequency and ISI independently influence the amplitude of the AV-induced M300 component.

Due to the existence of significant main effect, Bonferroni correction post hoc comparisons were conducted to identify differences between specific factor levels. The main effects of auditory frequency and ISI on M300 amplitude are illustrated in Fig 5. Bonferroni correction was applied separately within each post-hoc comparison family. For the frequency factor, pairwise comparisons revealed that the M300 amplitude at 6.3 kHz was significantly higher than that at 2.3 kHz ($p = 0.00002$, Cohen's d = 2.32) and 4.3 kHz ($p = 0.00011$, Cohen's d = 2.04). Similarly, 8.3 kHz also produced significantly higher responses than 2.3 kHz ($p = 0.000006$, Cohen's d = 2.72) and 4.3 khz ($p = 0.000069$, Cohen's d = 2.30). No significant difference was observed in 6.3 kHz-8.3 kHz ($p = 0.217$) and 2.3 kHz-4.3 kHz, suggesting a ceiling effect at higher frequencies.

For the ISI factor, the M300 amplitude at 7 s was significantly higher than that at 3 s ($p = 0.00005$, Cohen's d = 2.154) and 5 s ($p = 0.00061$, Cohen's d = 1.811). At 9 s, M300 amplitude was higher than 3 s ($p = 0.0004$, Cohen's d = 1.869) and 5 s ($p = 0.00437$, Cohen's d = 1.526). No significant difference was found between 7 s-9 s and 3 s-5 s.

The effect size values (Cohen's d > 1.1) across all significant comparisons support the presence of large between-group differences.

These results confirm that frequency and ISI parameters exert independent effects on AV-induced M300 activity, without interactive modulation. Increases in auditory frequency are associated with monotonic enhancement of M300 amplitude, whereas the ISI-dependent increase peaks at 7 s and stabilizes thereafter.

## Conclusion and discussion

This study is the first to non-invasively record audiovisual-induced ERMFs in a rat model using a high-sensitivity ($20 fT/\sqrt{Hz}$) SERF AM. By capturing superadditive responses in the rat brain, our results provide new insights into the mechanisms underlying audiovisual multisensory integration. The sensitivity, spatial resolution, and capability for cortical proximity measurements provided by the SERF AM enabled the detection of high signal-to-noise ratio ERMF signals under non-invasive conditions—something difficult to achieve in previous SQUID-based experiments.

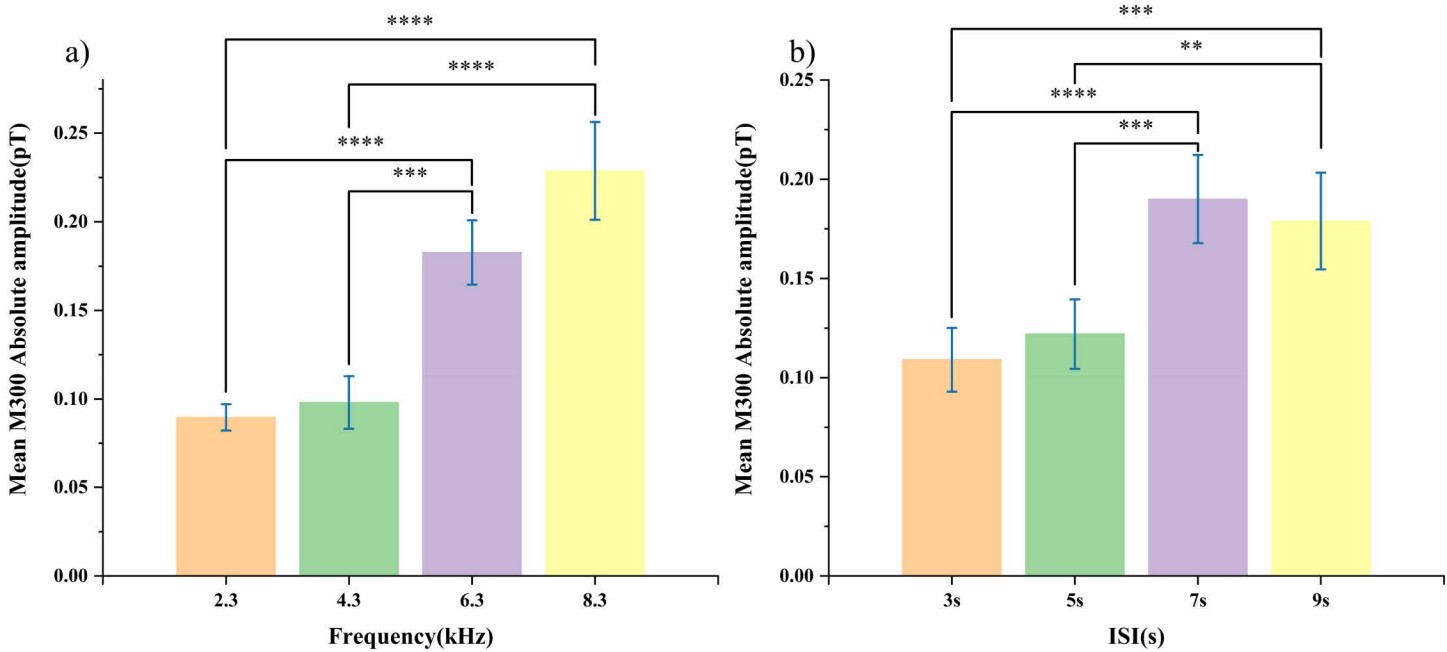

**Fig 5. (a) Bonferroni-adjusted pairwise comparison of M300 amplitude on frequency.** (b) Bonferroni-adjusted pairwise comparison of M300 amplitude on ISI. Main effects of frequency (a) and ISI (b) on AV-induced M300 amplitude. Error bars indicate ±SEM. Statistical significance between group pairs was determined by Bonferroni correction.

The superadditive neuromagnetic responses observed in this study is similar with nonlinear integration previously reported in human neuroimaging studies using MEG, EEG, and ERP techniques [38–40]. These studies demonstrate that audiovisual stimuli can evoke enhanced neural response that exceed the linear summation of unimodal responses, both in terms of response amplitude and spatial/temporal encoding precision. Our findings in rats further expands this understanding by systematically examining a range of stimulus frequencies and inter-stimulus intervals, which have not been thoroughly explored in human models, highlighting the utility of animal model for studying parameter-dependent modulation of multisensory processing.

The experimental results reveal that multisensory integration in rats is modulated by stimulus parameters, giving rise to both response enhancement and suppression. In the M300 component, AV stimulation under certain frequency–ISI combinations elicited ERMF amplitudes exceeding the linear sum of unimodal responses, consistent with superadditive effects. These effects were condition specific, and may be compatible with late-stage multisensory processing differences across stimulus parameters [37,41]. Meanwhile, during the early stage (M100), a general reduction of the visual channel responses was observed across most parameter combinations, with preservation only in select conditions. This observation may be consistent with early-stage selective modulation or competition between sensory channels.

Taken together, the coexistence of early suppression and late enhancement suggests that multisensory integration may differ across temporal stages of processing. One possible interpretation is that early responses are more susceptible to multichannel suppression, whereas later responses may reflect enhancement under multisensory stimulation. However, because the present study did not include behavioral measurements or concurrent electrophysiological validation, these interpretations should be regarded as provisional. More generally, the nonlinear dependence of the responses on frequency and ISI indicates that multisensory ERMF signals are sensitive to stimulus timing and spectral content under the present experimental conditions.

In interpreting these findings, it is worth noting that the influence of anesthesia on ERMF signals, particularly on the latency and amplitude of components such as M100 and M300, has not been comprehensively characterized in rodents. In addition, a fixed moving average smoothing window was used throughout the analysis. Although this approach helped suppress high frequency noise, its potential influence on the measured component amplitudes was not systematically evaluated in the current study. A more formal uncertainty analysis of SRI, such as bootstrap confidence intervals across animals, would also be valuable in future work with fully structured archived datasets. While sodium pentobarbital was employed in this study to minimize motion artifacts and maintain physiological stability, anesthesia may itself alter cortical excitability, response amplitude, and response latency. Future studies under awake or lightly anesthetized conditions may help further assess the robustness and specificity of the M300 responses observed here. Moreover, the current study included only adult female rats, primarily to reduce inter subject physiological variability, which may limit the generalizability of our findings across sexes. Subsequent research should consider incorporating male subjects or performing direct sex-based comparisons to explore potential gender differences in audiovisual integration.

Additionally, the non-invasive, high-sensitivity neuromagnetic recording methodology developed here may hold potential for future translational research, particularly for neurodevelopmental disorders. Although behavioral or disease-specific data were not included in the current study, future research may explore the application of this approach to characterize abnormal multisensory dynamics in translational models of neurodevelopmental disorders, contributing to long-term efforts toward early screening and monitoring.

## Supporting information

**S1 Data. Dataset and code.**
(ZIP)

## Acknowledgments

We gratefully thank all members of our research group for their valuable contributions and efforts throughout this study.

## Author contributions

**Conceptualization:** Jia Yao, Yi Ruan.

**Data curation:** Kexun Tang.

**Formal analysis:** Kexun Tang.

**Investigation:** Guanzhong Lu.

**Methodology:** Guanzhong Lu.

**Resources:** Yi Ruan, Qiang Lin.

**Supervision:** Jia Yao, Yi Ruan, Qiang Lin.

**Validation:** Peixin Zhang, Hao Wang, Zixuan Wang.

**Writing – original draft:** Kexun Tang.

**Writing – review & editing:** Kexun Tang, Yi Ruan.

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
