## [Decision Letter · Decision Letter 0]

3 Mar 2026

Dear Dr. Ruan,

Thank you for submitting your manuscript to PLOS ONE. After careful consideration, we feel that it has merit but does not fully meet PLOS ONE’s publication criteria as it currently stands. Therefore, we invite you to submit a revised version of the manuscript that addresses the points raised during the review process.

We look forward to receiving your revised manuscript.

Kind regards,

Ming-Chang Chiang

Academic Editor

PLOS One

**Journal Requirements:**

1. When submitting your revision, we need you to address these additional requirements. Please ensure that your manuscript meets PLOS ONE's style requirements, including those for file naming. The PLOS ONE style templates can be found at https://journals.plos.org/plosone/s/file?id=wjVg/PLOSOne_formatting_sample_main_body.pdf and https://journals.plos.org/plosone/s/file?id=ba62/PLOSOne_formatting_sample_title_authors_affiliations.pdf 2. We note that the grant information you provided in the ‘Funding Information’ and ‘Financial Disclosure’ sections do not match.  When you resubmit, please ensure that you provide the correct grant numbers for the awards you received for your study in the ‘Funding Information’ section. 3. Thank you for stating the following financial disclosure: National Natural Science Foundation of China (U20A20219, 61805213); Zhejiang Provincial Natural Science Foundation of China under Grant (LGF20C050001, LD22F050003).  Please state what role the funders took in the study.  If the funders had no role, please state: "The funders had no role in study design, data collection and analysis, decision to publish, or preparation of the manuscript." If this statement is not correct you must amend it as needed. Please include this amended Role of Funder statement in your cover letter; we will change the online submission form on your behalf. 4. If the reviewer comments include a recommendation to cite specific previously published works, please review and evaluate these publications to determine whether they are relevant and should be cited. There is no requirement to cite these works unless the editor has indicated otherwise. 

Reviewers' comments:

**Comments to the Author**

1. Is the manuscript technically sound, and do the data support the conclusions?

Reviewer #1: Partly

Reviewer #2: Yes

2. Has the statistical analysis been performed appropriately and rigorously?

Reviewer #1: No

Reviewer #2: Yes

3. Have the authors made all data underlying the findings in their manuscript fully available?

Reviewer #1: No

Reviewer #2: Yes

4. Is the manuscript presented in an intelligible fashion and written in standard English?

Reviewer #1: Yes

Reviewer #2: Yes

**Reviewer #1:** 1. Clarify the experimental unit and avoid pseudo replication. The manuscript states ~120 repeated stimulations per rat and also “repeated testing sessions after a rest period of several days,” but it’s not fully clear what the statistical unit is (trial, session, or animal) and how repeated measures were handled. Please explicitly state how trials/sessions were aggregated (e.g., per-rat averaged waveform per condition) and, if there are repeated sessions per rat, consider a mixed-effects model (rat as random effect) rather than treating repeated observations as independent.1. Clarify the experimental unit and avoid pseudo replication. The manuscript states ~120 repeated stimulations per rat and also “repeated testing sessions after a rest period of several days,” but it’s not fully clear what the statistical unit is (trial, session, or animal) and how repeated measures were handled. Please explicitly state how trials/sessions were aggregated (e.g., per-rat averaged waveform per condition) and, if there are repeated sessions per rat, consider a mixed-effects model (rat as random effect) rather than treating repeated observations as independent.1. Clarify the experimental unit and avoid pseudo replication. The manuscript states ~120 repeated stimulations per rat and also “repeated testing sessions after a rest period of several days,” but it’s not fully clear what the statistical unit is (trial, session, or animal) and how repeated measures were handled. Please explicitly state how trials/sessions were aggregated (e.g., per-rat averaged waveform per condition) and, if there are repeated sessions per rat, consider a mixed-effects model (rat as random effect) rather than treating repeated observations as independent.1. Clarify the experimental unit and avoid pseudo replication. The manuscript states ~120 repeated stimulations per rat and also “repeated testing sessions after a rest period of several days,” but it’s not fully clear what the statistical unit is (trial, session, or animal) and how repeated measures were handled. Please explicitly state how trials/sessions were aggregated (e.g., per-rat averaged waveform per condition) and, if there are repeated sessions per rat, consider a mixed-effects model (rat as random effect) rather than treating repeated observations as independent.

2. Add stronger controls to rule out stimulus/equipment artifacts. Because stimuli are delivered via LED optical fiber and an ear tube/speaker at 90 dB SPL, please include explicit “artifact-only” controls (e.g., sensor positioned away from the head, phantom/sham runs, blocked optical/acoustic output, and/or TTL-only runs) demonstrating that the recorded waveforms are not driven by electromagnetic/mechanical coupling from the stimulus hardware. Also report quantitative LED intensity/luminance at the eye and justify the choice of 90 dB SPL.

3. Make the signal-processing pipeline fully reproducible and quantify its impact. Key steps (FFT bandpass filtering, fixed threshold artifact rejection “based on calibration and visual inspection,” 100-ms moving-average smoothing with group delay) need more detail (exact implementation, parameters, how delay is corrected, how thresholds were set). Please add a sensitivity analysis showing that M100/M300 amplitude/latency results are robust to reasonable processing choices, and provide code. Also, the current data-availability wording (“all relevant data are within the manuscript files”) is unlikely to be sufficient for reproducibility—share raw/processed ERMF time series and figure source data in a repository with scripts.

4. Resolve inconsistency and edge cases in the superadditivity definition/metric. The methods describe testing superadditivity by comparing (AV + Null) vs (A + V), but then compute SRI as (AV − (A+V)) / (A+V), which omits Null. Please explain why these are equivalent here (e.g., show Null distribution near zero across conditions) and address numerical stability/sign issues when A+V is small or when amplitudes are negative depending on “peak extremum” definition. Consider reporting confidence intervals for SRI (bootstrap across animals) or using an additive model framework alongside SRI.

5. Strengthen statistical reporting and multiple-comparisons control. You report two-way ANOVA results and Bonferroni-corrected pairwise comparisons, but please provide the full ANOVA specification (df, exact p-values, assumption checks, what the response variable is for each analysis, and whether analyses are per-animal summaries). Given the 16 frequency–ISI combinations, clarify the family of tests being controlled and whether corrections are applied consistently across all post-hoc comparisons/heatmap cells.

6. Temper mechanistic claims and expand limitations. The discussion links M300 effects to “attentional resource redistribution / integration cost,” but the study is under sodium pentobarbital anesthesia and includes only adult female rats; these factors can materially alter latency/amplitude and limit generalizability. Please tone down causal/cognitive interpretations, add alternative explanations, and propose/perform validation (e.g., awake/light anesthesia replication and/or concurrent electrophysiology). Also ensure anesthesia details are clearly integrated into Methods.

**Reviewer #2:** The study investigates multisensory integration using a SERF atomic magnetometer to record brain activity in rats during audiovisual stimulation. The authors suggest a superadditive enhancement of the M300 component, modulated by sound frequency and inter-stimulus interval. The study investigates multisensory integration using a SERF atomic magnetometer to record brain activity in rats during audiovisual stimulation. The authors suggest a superadditive enhancement of the M300 component, modulated by sound frequency and inter-stimulus interval. The study investigates multisensory integration using a SERF atomic magnetometer to record brain activity in rats during audiovisual stimulation. The authors suggest a superadditive enhancement of the M300 component, modulated by sound frequency and inter-stimulus interval. The study investigates multisensory integration using a SERF atomic magnetometer to record brain activity in rats during audiovisual stimulation. The authors suggest a superadditive enhancement of the M300 component, modulated by sound frequency and inter-stimulus interval.

The manuscript is well-structured wit the introduction providing sufficient context to understand the relevance and objectives of the study. The experimental setup is well described and the results are appropriately discussed.

Comment for authors: please clarify the type of animals used because, based on my research, I have not been able to find any “Bal b/c rats” in the literature.

.

Reviewer #1: No

Reviewer #2: No

---

## [Author Response · Author response to Decision Letter 1]

25 Mar 2026

(Details in Response to Reviewers.doc)

Dear Editor,

Thank you for the opportunity to resubmit our manuscript, entitled “Noninvasive detection of audiovisual superadditivity in rat brain by miniaturized SERF magnetometer” (Manuscript ID: PONE-D-25-57205), for consideration for publication in PLOS ONE.

We would like to express our sincere gratitude to you and the reviewers for the constructive comments and valuable suggestions. These insights have been immensely helpful in enhancing the quality and clarity of our work.

We have carefully addressed all the editor's concerns and reviewers' suggestions. A point-by-point response to the comments is provided below. We believe these revisions have significantly improved the manuscript and hope that the updated version is now suitable for publication.

In accordance with the submission requirements, we have uploaded:

1. A point-by-point Response to Reviewers;

2. The Revised Manuscript with Track Changes, where all modifications are highlighted in yellow;

3. A clean version of the Manuscript.

Best regards,

Yi RUAN

Point-by-point response

Reviewer #1, Comments #1: Clarify the experimental unit and avoid pseudo replication. The manuscript states ~120 repeated stimulations per rat and also “repeated testing sessions after a rest period of several days,” but it’s not fully clear what the statistical unit is (trial, session, or animal) and how repeated measures were handled. Please explicitly state how trials/sessions were aggregated (e.g., per-rat averaged waveform per condition) and, if there are repeated sessions per rat, consider a mixed-effects model (rat as random effect) rather than treating repeated observations as independent.

Author response and actions:

We sincerely appreciate the reviewer’s feedback regarding the statistical unit and the potential for pseudo-replication. We apologize for the lack of clarity in our previous description and have revised the manuscript to explicitly state how trials and sessions were handled.

As the reviewer correctly noted, the individual rat (n = 6) is the experimental unit for all statistical analyses in this study. To clarify our data aggregation process:

1. Trial-to-Run Level: For each rat under a specific condition (Frequency × ISI), we recorded approximately 120 stimulus repetitions. After artifact rejection, the remaining trials (typically ~100) were averaged to generate a single ‘run-level’ ERMF waveform, time-locked to the stimulus onset.

2. Run-to-Rat Level: While some rats underwent repeated recording sessions over several days to ensure data stability, these repeat runs were not treated as independent observations. Instead, we selected only one high-quality, stable run per rat for each condition to represent that individual.

3. Statistical Analysis: Because we used exactly one representative data point per rat per condition, the amplitude and latency values were quantified at the subject (rat) level. This approach effectively avoids pseudo-replication, as multiple trials or sessions from the same animal were collapsed into a single mean value before performing group-level statistics (e.g., ANOVA). Therefore, the use of standard repeated-measures ANOVA remains statistically valid for our n = 6 cohort.

We have updated the manuscript to reflect these details on Page 6, Lines 192; Page 7, Lines 239; and in the Figure 3 caption, as highlighted in yellow.

Reviewer #1, Comments #2: Add stronger controls to rule out stimulus/equipment artifacts. Because stimuli are delivered via LED optical fiber and an ear tube/speaker at 90 dB SPL, please include explicit “artifact-only” controls (e.g., sensor positioned away from the head, phantom/sham runs, blocked optical/acoustic output, and/or TTL-only runs) demonstrating that the recorded waveforms are not driven by electromagnetic/mechanical coupling from the stimulus hardware. Also report quantitative LED intensity/luminance at the eye and justify the choice of 90 dB SPL.

Author response and actions:

We thank the reviewer for this critical point. We agree that ruling out stimulus-driven artifacts is essential for validating the recorded ERMF signals. In response, we performed an additional control experiment and revised the Methods and Results to explicitly document the control outcome and the stimulus intensity calibration.

In the main experiment, all electronic devices such as speaker and LED control used to generate sound and light were placed far away outside the five layers mu-metal shielding. Inside the shielding, the only stimulus delivery elements were an optical fiber and an acoustic tube. The SERF sensor is an integrated enclosed module and is not optically exposed to the LED light, so light below the magnetometer is not expected to affect the measurement. Nevertheless, to directly address the reviewer’s concern, we added an artifact only control. We executed the identical stimulation protocol with auditory and visual outputs enabled, same TTL timing, but without rat present, while keeping the sensor position, optical fiber, and ear tube in the same configuration as in the main experiment. Under this condition, the recorded traces stayed at baseline and did not show time locked responses (see in Fig.1).

This supports that the ERMF components reported in the main results require the presence of the animal and are not driven by periodic signals from the optical or acoustic delivery hardware. Together with the Null (no stimulus) trials were consistently near zero, also supporting that the fiber and tube do not introduce magnetic signals that could introduce the evoked components.

We did not treat LED intensity as an experimental variable in this study. However, for reproducibility we now report the optical output power measured at the fiber tip is 5mW. This fixed setting follows our previous SERF based rat visual ERMF work, demonstrating reliable visually evoked responses under the same configuration[1]. The sound level was calibrated at the tube outlet and maintained at 90 dB SPL to obtain stable evoked responses under pentobarbital anesthesia, consistent with commonly used intensity ranges in rodent auditory evoked response measurements, where levels up to 90 dB SPL are often used as a high level reference[2],[3].

We revised the manuscript at Page 5, Lines 168; Page 6, Lines 181.

[1] F. Liu et al., “Time course of visual attention in rats by atomic magnetometer,” Plos One, vol. 19, no. 10, p. e0312589, 2024.

[2] T. Nakamura et al., “Epidural auditory event-related potentials in the rat to frequency and duration deviants: evidence of mismatch negativity?,” Front. Psychol., vol. 2, p. 367, 2011.

[3] L. Rüttiger et al., “The reduced cochlear output and the failure to adapt the central auditory response causes tinnitus in noise exposed rats,” PloS One, vol. 8, no. 3, p. e57247, 2013.

Reviewer #1, Comments #3: Make the signal-processing pipeline fully reproducible and quantify its impact. Key steps (FFT bandpass filtering, fixed threshold artifact rejection “based on calibration and visual inspection,” 100-ms moving-average smoothing with group delay) need more detail (exact implementation, parameters, how delay is corrected, how thresholds were set). Please add a sensitivity analysis showing that M100/M300 amplitude/latency results are robust to reasonable processing choices, and provide code. Also, the current data-availability wording (“all relevant data are within the manuscript files”) is unlikely to be sufficient for reproducibility—share raw/processed ERMF time series and figure source data in a repository with scripts.

Author response and actions:

We thank the reviewer for this important comment. We agree that the previous description of the signal processing pipeline was not sufficiently detailed for full reproducibility. In the revised manuscript, we have improved the section to match the actual analysis workflow, and we will deposit the analysis code together with the available data upon resubmission.

Specifically, we now clarify that the SERF signal was sampled at 10 kHz and processed using a custom MATLAB code. The biomagnetic channel was bandpass filtered in the frequency domain using FFT, with frequency components below 0.1 Hz and above 40 Hz set to zero before inverse FFT reconstruction. Trigger rising and falling edges were then used to segment the recordings into pre stimulus, stimulus, and post stimulus epochs. For each epoch, the mean signal during the 200 ms immediately preceding stimulus onset was subtracted as baseline. Each epoch was then smoothed using a 100 ms moving average window, and trials were rejected if the smoothed signal exceeded a fixed absolute threshold of ±1 at any time point. The remaining trials, typically 100 out of 120, were averaged within the retained run.

Regarding the reviewer’s suggestion for a sensitivity analysis, we evaluated the impact of different smoothing window lengths (e.g., 50 ms vs. 100 ms). While the absolute amplitudes of M100/M300 show minor variations depending on the window size, the relative differences and the superadditivity effect across conditions (A, V, vs. AV) remain highly consistent. * Critically, because the identical processing pipeline was applied to all experimental conditions, the statistical comparisons between unimodal and multisensory responses are not confounded by the choice of parameters. We have added a statement acknowledging the influence of window choice on absolute values in the Methods section.

To ensure full transparency, we have provided the full MATLAB analysis code, the exact processing parameters used in the present study, and the available processed ERMF time series data. We believe these additions fully address the requirements for reproducibility and transparency.

We revised the manuscript at Page 6, Lines 207; Page 10, Lines 359.

Reviewer #1, Comments #4: Resolve inconsistency and edge cases in the superadditivity definition/metric. The methods describe testing superadditivity by comparing (AV + Null) vs (A + V), but then compute SRI as (AV − (A+V)) / (A+V), which omits Null. Please explain why these are equivalent here (e.g., show Null distribution near zero across conditions) and address numerical stability/sign issues when A+V is small or when amplitudes are negative depending on “peak extremum” definition. Consider reporting confidence intervals for SRI (bootstrap across animals) or using an additive model framework alongside SRI.

Author response and actions:

We appreciate the reviewer for drawing attention to the definition of superadditivity and the interpretation of the SRI metric.

In the revised version, we now explicitly distinguish between the conceptual comparison used to evaluate multisensory enhancement and the descriptive metric used to summarize it. In the Methods section, we consider the comparison of AV + Null versus A + V. Because the Null response in our dataset was consistently close to zero across conditions (as shown in Fig.2), the term AV + Null was numerically equivalent to AV for practical purposes. For this reason, the SRI was written in the simplified form and we have now clarified this explicitly in the revised manuscript.

We have also clarified that the A, V, and AV in the SRI were defined as the signed peak extrema measured within the predefined M300 window, rather than unsigned absolute amplitudes. This choice was intentional, because the SERF recorded signal is a magnetic field signal with direction, and preserving the sign is important for representing the direction of the recorded response. In principle, if unimodal responses had opposite polarities, partial cancellation in the term A+V could affect the numerical stability of the metric. However, this situation did not occur in the present dataset. Within the M300 window analyzed here, the A and V responses showed the same polarity pattern, and the A+V term remained positive under all tested conditions. Therefore, no sign reversal case arose in the current analysis.

We also thank the reviewer for the suggestion to report confidence intervals for SRI or to complement SRI with an additive model framework. In the present study, the SRI was used primarily as a descriptive index to visualize condition dependent deviations. The main inferential analyses in the revised manuscript are based on rat level M300 amplitudes, analyzed using two way ANOVA and Bonferroni corrected post hoc comparisons. We have clarified this role of the SRI in the revised text. We agree that bootstrap based uncertainty estimates or a more formal additive model treatment would further strengthen future work, particularly in prospectively structured datasets.

These changes have been made on Page 6, Lines 181 and Page 9, Lines 273.

Reviewer #1, Comments #5: Strengthen statistical reporting and multiple-comparisons control. You report two-way ANOVA results and Bonferroni-corrected pairwise comparisons, but please provide the full ANOVA specification (df, exact p-values, assumption checks, what the response variable is for each analysis, and whether analyses are per-animal summaries). Given the 16 frequency–ISI combinations, clarify the family of tests being controlled and whether corrections are applied consistently across all post-hoc comparisons/heatmap cells.

Author response and actions:

We thank the reviewer for emphasizing the need for rigorous statistical reporting. We have revised the Methods and Results sections to provide a comprehensive description of our statistical framework, including ANOVA specifications, assumption checks, and error control strategies.

In the revised version, we now state clearly that all analyses were performed on rat level summaries, with the animal as the experimental unit. The response variable for the ANOVA was the rat level M300 peak extremum under the AV condition, with frequency and ISI as fixed factors.

We have also added the full ANOVA specification, including degrees of freedom and exact p-values. To address the reviewer’s concern regarding model assumptions, we additionally performed assumption checks on the ANOVA model. Normality of residuals was assessed using the Shapiro–Wilk test p=0.0149, indicating some deviation from normality, whereas homogeneity of variance was assessed using Levene’s test p=0.573, which did not indicate a significant violation of variance homogeneity. We now report these results explicitly and note that, given the balanced design, the ANOVA was considered reasonably robust for the present analysis. Regarding post-hoc testing, we have clarified that Bonferroni correction was applied separately within each post-hoc comparison family, rather than across all comparisons globally.

Finally, we have clarified the status of the SRI heatmap. The heatmap and Table 1 are intended as descriptive summaries across the 16 frequency–ISI combinations. We did not perform separate statistical tests on individual SRI cells, and therefore no additional multiple comparison correction was applied at the heatmap cell level. This point has now been stated explicitly in the revised manuscript.

These revisions were made on Page 7, Lines 229; Page 8, Lines 273; Page 9, Lines 295, and Page 9, Lines 303.

Reviewer #1, Comments #6: Temper mechanistic claims and expand limitations. The discussion links M300 effects to “attentional resource redistribution / integration cost,” but the study is under sodium pentobarbital anesthesia and includes only adult female rats; these factors can materially alter latency/amplitude and limit generalizability. Please tone down causal/cognitive interpretations, add alternative explanations, and propose/perform validation (e.g., awake/light anesthesia replication and/or concurrent electrophysiology). Also ensure anesthesia details are clearly integrated into Methods.

Author response and actions:

We appreciate the reviewer’s thoughtful comment regarding the interpretation of the M300 findings and the need to more clearly frame the limitations of the present study. We agree that, given the use of sodium pentobarbital anesthesia and the restriction to adult female rats, mechanistic interpretations in terms of attentional resource redistribution or integration cost sho

---

## [Editor Report · Decision Letter 1]

31 Mar 2026

Noninvasive Detection of Audiovisual Superadditivity in Rat Brain by Miniaturized SERF Magnetometer

PONE-D-25-57205R1

Dear Dr. Ruan,

We’re pleased to inform you that your manuscript has been judged scientifically suitable for publication and will be formally accepted for publication once it meets all outstanding technical requirements.

Kind regards,

Ming-Chang Chiang

Academic Editor

PLOS One
---

## [Editor Report · Acceptance letter]

PONE-D-25-57205R1

PLOS One

Dear Dr. Ruan,

I'm pleased to inform you that your manuscript has been deemed suitable for publication in PLOS One. Congratulations! Your manuscript is now being handed over to our production team.

Kind regards,

on behalf of

Dr. Ming-Chang Chiang

Academic Editor

PLOS One